# Superior pedal function recovery of newly designed three spike insole over total contact insole in refractory plantar fasciitis: A randomized, double-blinded, non-inferiority study

**Dong Woo Shim**[1], **Seung-Yong Sung**[1], **Whan-Yong Chung**[1], **Kyung-Yil Kang**[2], **Sung-Jun Park**[3], **Jin Woo Lee**[4], **Dong Sik Chae**[1] *

1 Department of Orthopedic Surgery, International St. Mary's Hospital, Catholic Kwandong University College of Medicine, Incheon, South Korea, 2 Department of Medicine, Catholic Kwandong Graduate School, Gangneungsi, Gangwondo, South Korea, 3 School of Mechanical, Automotive and Aeronautical Engineering Korea National University of Transportation, Chungju, Chungbuk, South Korea, 4 Department of Orthopaedic Surgery, Yonsei University College of Medicine, Seoul, South Korea

* drchaeos@gmail.com

**Data Availability Statement:** All relevant data are within the paper and its Supporting information files.

## Abstract

### Introduction

Plantar fasciitis is one of the common foot complaints that is chronic and can induce dysfunction. Total contact insole (TCI) is simple but effective in treating plantar fasciitis. Despite its effect, the cost and long duration for production have been the major flaws. Therefore, we developed a newly designed three-spike insole (TSI) that can be commercially productive and compared its clinical outcomes to TCI.

### Methods

Patients with plantar fasciitis refractory to conservative treatment for more than 6 weeks were candidates. We produced insoles with hardness of 58 ± 5 Shore-A. Twenty-eight patients were randomized with equal allocation to either TSI or TCI. The following assessment tools were used: visual analog scale (VAS), American Orthopaedic Foot and Ankle score, Foot and Ankle Outcome Score, Karlsson-Peterson (KP) score, Short Form-36 for quality of life, and Foot Function Index. Non-inferiority was declared if VAS was within the statistical variability of minimal important difference. A blinded assessor evaluated the groups at baseline and after 6, 12, and 24 weeks.

### Results

The groups were homogenous for majority of variables at baseline. Overall patient-reported satisfaction showed improvement from mean 5.2 (range, 1–12) weeks of wearing and all clinical outcome scores showed significant improvements in both groups over time on Friedman test (p ≤.032). TSI showed non-inferiority to TCI at each time point. Post hoc analysis

**Funding:** DWS This research was supported by Basic Science Research Program through the National Research Foundation of Korea (NRF) funded by the Ministry of Science, ICT & Future Planning(NRF-2019R1G1A1098885). https://ernd.nrf.re.kr/index.do The funders had no role in study design, data collection and analysis, decision to publish, or preparation of the manuscript.

**Competing interests:** The authors have declared that no competing interests exist.

revealed that many scales showed significant superiority of TSI at 3 month (p $\leq$ .008) and KP score at 6 month (p < .001).

## Conclusion

We reaffirmed that semi-rigid insole is effective in refractory plantar fasciitis and showed TSI restores pedal function more rapidly than TCI. TSI can be not only effective in deriving better clinical outcomes but also be manufactured for popularization to lower the price and producing time of orthosis.

## Introduction

Plantar fasciitis is one of the common foot complaints that is chronic and can induce dysfunction. Approximately 10% of population develops heel pain during lifetime [1] Distinguishing feature is the starting heel pain on the first step in the morning and it can wax and wane during daytime. Approximately 90% of patients improve in a year with conservative management such as stretching exercise, night splint, orthotics, extracorporeal shockwave therapy, and local injection [2–8] These procedures are generally based on outpatient clinic and cost about 23.4 million dollars a year in Korea according to National Health Insurance Service in 2018. Recalcitrant plantar fasciitis, unresponsive to conservative care, can be treated with surgical intervention but the reports vary upon the outcomes [9–12].

Foot orthosis is simple but effective in treating plantar fasciitis. It can be beneficial to patients in that population vulnerable to plantar fasciitis who tend to have a job with a long time walking or standing and not have enough time spent on treatment including stretching exercise or visiting a clinician [13] Many types of orthoses have been introduced but the efficacy are debatable. Some authors reported that prefabricated orthoses are effective enough compared to the customized foot orthoses [14, 15] However, majority of the studies showed superiority of the customized foot orthoses in the form of total contact insole to the prefabricated and sham orthoses [16–21] Despite the effect, the cost and long duration for production have been its major flaws. On average, it takes a week and costs 150 dollars for production.

Previous biomechanical studies have proposed that the medial surface of orthosis should stabilize the apical bony structure to effectively support the medial longitudinal arch. The studies have also proposed that the optimal insole contact surface should be over 54% of the plantar fascia to lessen the fascial stress and peak pressure [22, 23] Furthermore, structural analysis tells us that bending moments are higher in concentrated load model than in uniformly distributed load model [24] Therefore, we developed newly designed insole that can be commercially productive, satisfying the above mentioned conditions. We hypothesized that the newly designed insole would not be inferior to the existing total contact insole and can have the merit to be manufactured for popularization due to its reproducible design.

## Methods

### Study design

A parallel double-blinded randomized controlled study with a 6-month follow-up duration was carried out from February 2019 to April 2020. This study was approved by our institution. Prior to inclusion, all participants voluntarily participated in this study and gave written informed consent; and rights of participants were protected. Patients over 19-year-old with plantar fasciitis refractory to conservative treatment including stretching, resting, ice bag, and

extracorporeal shock wave treatment (ESWT) etc. other than orthotics for more than 6 weeks were candidates for this study. Furthermore, patients with visual analog scale (VAS) of 5 or higher pain when taking the first step were included. Exclusion criteria were previous insole use, injection around the foot and ankle within 3 months, pulselessness of dorsalis pedis and posterior tibial artery, other systemic inflammatory diseases, comorbidities that can mimic symptom such as Achilles tendinitis and subtalar arthritis, history of foot and ankle fracture, and hindfoot deformity. This study was registered on the clinical trials registry system and the manuscript was prepared according to the CONSORT statement. This study was approved by the Catholic Kwandong University International St. Mary's Institutional Review Board (IS18OISI0072) and registered in Clinical Research Information Service (KCT0004737, first registration (14/02/2020)). We retrospectively registered this study because we did not know the existence and necessity of the public registration. However, the authors confirm that all ongoing and related trials for this intervention are registered. All methods were carried out in accordance with relevant guidelines and regulations.

Twenty-eight patients were enrolled from the outpatient clinic and informed consents were obtained. An excel-generated four block randomization list, which was created by a clinician not involved in this trial for concealment, was used to randomly allocate patients to the newly designed three-spike insole (TSI) group or total contact insole (TCI) group. A blinded physician assessed the outcome data.

## Sample size

The sample size was calculated for a non-inferiority trial of 2 independent means between the groups using VAS as the main variable. For determination of the minimal important difference (MID) of 2-cm on a 10-cm pain VAS, a 5% alpha set, 20% beta error, and standard deviation of 1.9 cm were established [19, 25] Twelve participants in each group were required for a minimal sample size; it was determined that each group consists of 14 patients, considering a possible loss of 10%.

## Intervention

We used a 3-dimensional (3D) printer (Mandme, [R]PLABS) to produce an insole. Both feet of all patients were copied with Copyfoam and the copies were sent to a manufacturer. All insoles were manufactured from thermoplastic polyurethane (TPU, density 12 g/cm$^3$) and had a 58 ± 5 Shore-A hardness filling up to 30% of the insole with TPU. Newly designed insole had three hemispheric spikes along the longitudinal medial arch starting from the center of plantar fascia. If a person's foot was 28.5 cm long, the measured length of plantar fascia in Copyfoam was about 12 cm. Therefore, each spike was designed to have a 2 cm diameter to support approximately 50% of the medial longitudinal arch; height of the first and third spike was 0.8 cm and that of the middle one was 1.2 cm. The first and the third spikes were placed at the most distal and proximal points of the medial longitudinal arch that were thought to be appropriate for stimulating the fascia based on foot posture on Copyfoam. The distance between the spikes differed among the patients depending on the length of the patient's arch. Based on the initial copied foam, the total contact insole was manufactured to have structures supporting the arch on its medial border like the general customized total contact orthosis (Fig 1). Bending moments of three estimated points in two insoles were calculated using structural mechanics.

## Clinical evaluation

Symptoms, pedal function, and foot and ankle related quality of life were the target evaluation. Clinical outcomes were assessed using maximal pain VAS in a day, American Orthopaedic

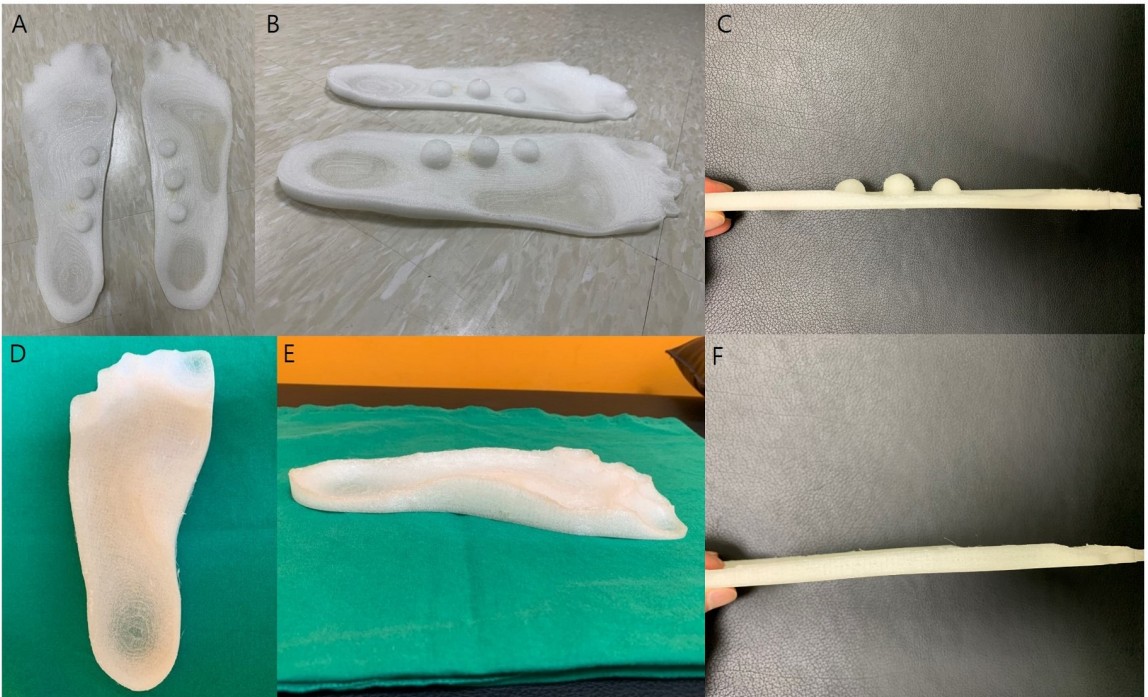

**Fig 1. A-C) Three spike insole (TSI) and D-F) total contact insole (TCI).** At baseline, all participants were given either of the above-mentioned insoles according to the randomization for daily use for 6 months. They were instructed to wear it on shoes or slippers as much as they could in a day and were not allowed to have more intervention including stretching exercise during the research. All participants were assessed by a blinded physician at baseline, 6 weeks, 3 months, and 6 months after wearing the insole.

Foot and Ankle Society (AOFAS) score, Foot and Ankle Outcome Score (FAOS), Karlsson-Peterson (KP) score, Medical Outcomes Study Short Form-36 (SF-36) for quality of life, and Foot Function Index (FFI).

## Statistical analysis

An intention to treat analysis was used—for the patients who were dropped out from further evaluations, the last data collected were used in the subsequent evaluations. Patient demographics and clinical outcomes are presented as mean ± SD or count (percentage). We used Mann-Whitney test for quantitative variables and Fisher's exact test for categorical variables for comparison between the groups. Non-inferiority test was used to assess the VAS between 2 groups. Friedman test was used for comparison between two groups over time. Post hoc analysis with Mann-Whitney test was conducted with a Bonferroni's correction, resulting in a significance level set at $p < .0125$. The level of statistical significance was set at $p < .05$. The Statistical Package for the Social Sciences (SPSS, version 21.0, IBM Corp., Armonk, NY, USA) was used for all statistical analyses. Non-inferiority test was used to investigate if the newly designed insole would not be inferior to the existing total contact insole.

## Results

Twenty-eight patients were randomized to two groups with equal probability; a CONSORT flowchart is shown in Fig 2. Three participants were dropped from the study because of insole inconvenience and 1 participant was dropped for the treatment of radiating pain from herniated lumbar disc. Demographic characteristics are summarized in Table 1. The symptom

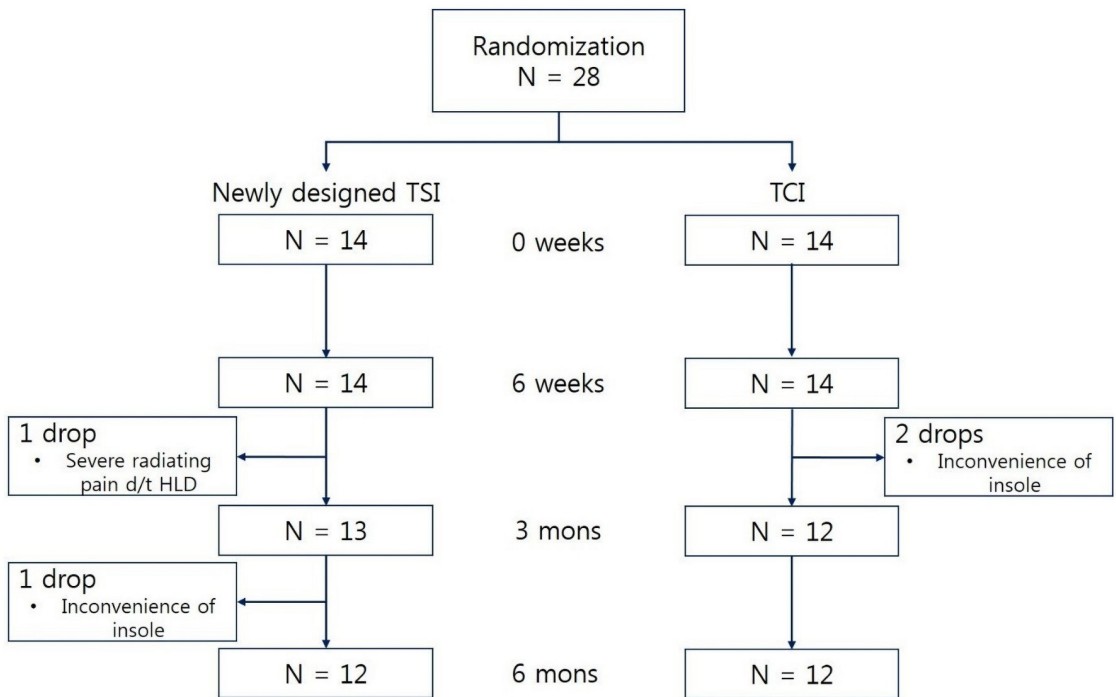

**Fig 2. CONSORT flowchart.** Twenty-eight participants were distributed to 2 groups and 4 participants were lost to follow-up thoroughly. Finally, twelve participants were evaluated in each group.

duration and previous treatment duration tended to be longer in TSI group but were not significant. Mean age of all participants was 48.2 ± 9.4 (range, 28–67), mean insole usage time was 4.0 ± 2.9 (range, 2.5–12.0) hours a day, and mean time to clinical improvement was 5.2 ± 2.6 (range, 1–12) weeks in all groups. Most common previous treatments were medication and ESWT. All participants took medicine for more than a month or underwent ESWT for more than 3 times; and 26 participants received both treatments. Three patients had previous local injection history around the plantar fascia for more than 3 months before enrollment.

**Table 1. Patient demographics of newly designed TSI and TCI Groups.**

|  | TSI (N = 14) | TCI (N = 14) | P value |
|---|---|---|---|
| Age (years) | 48.2 ± 9.9 | 48.2 ± 8.8 | .999 |
| Male gender | 4 (28.6%) | 4 (28.6%) | .999 |
| Side (right: left: both) | 5: 2: 7 | 6: 3: 5 | .663 |
| Height (cm) | 162.9 ± 6.9 | 162.5 ± 8.2 | .787 |
| BMI (kg/m$^2$) | 25.2 ± 2.6 | 25.1 ± 2.7 | .965 |
| Symptom duration (months) | 25.3 ± 18.7 | 16.3 ± 11.5 | .173 |
| Previous treatment duration (months) | 20.1 ± 19.9 | 8.4 ± 2.9 | .068 |
| Insole usage time (hour/day) | 4.8 ± 2.6 | 3.4 ± 3.0 | .875 |
| Time to feel improvement with insole (weeks) | 5.6 ± 2.2 | 4.8 ± 2.8 | .630 |

TSI, three spike insole; TCI, total contact insole; BMI, body mass index.

## Bending moments

The bending moments of two insoles are drawn in Fig 3. When we assume the plantar fascia as a bow string, TCI can evenly support the fascial longitudinal arch with a uniformly distributed load. On this wise, bending moments added on plantar fascia at some point in the model can be shown as Fig 3A [24]. Otherwise, TSI can support and stimulate the longitudinal arch at the specified 3 points as shown in Fig 3B. Bending moments at each point in this several concentrated load model tend to be bigger than those of the uniformly distributed load model.

## Non-inferiority analysis

Mean differences of VAS with 95% confidential interval (CI) at each time point of 6 weeks, 3 months, and 6 months were 0.6 (95% CI 0.6–2.6), 0.8 (95% CI -1.7–0.9), and 0.4 (95% CI -1.6–0.6), respectively. All value differences except for that of 6 weeks were within MID. The scale at 6 weeks showed superiority of TSI. These indicate that the TSI shows non-inferiority to TCI according to VAS.

## Clinical outcomes

Almost all parameters except for KP score, Sport/Rec subscale in FAOS, and PCS in SF-36 showed significant improvement at 6 weeks in all participants (p ≤.027). Comparison between the groups are shown in Table 2 and Fig 4. All clinical outcome scores showed significant improvements over time in both groups on Friedman test (p ≤.032). Post hoc analysis with Mann-Whitney test conducted with Bonferroni correction further revealed that there were no significant differences of variables between the groups at baseline and 6 weeks; but many scales showed a notable significant superiority of TSI at 3 months (p ≤.008). Furthermore, KP score at 6 months was superior in TSI group (p < .001). We can conclude that the TSI group showed significantly superior results in all clinical outcomes to TCI group over time during 6 months of follow up. There was no adverse event in participants.

## Discussion

This study showed that a newly designed TSI showed non-inferiority to conventional TCI according to VAS and a relatively rapid functional improvement during the 6-month follow up. Both types of insoles were effective enough to reduce pain and increase pedal function and foot-related quality of life at mean 5.2 weeks of put on. Pain scale did not show significant differences at every time point between the groups and value differences of VAS were within MID. Moreover, KP score, FFI, pain and symptom subscales in FAOS, and SF-36 scores at 3 months were significantly superior in TSI group compared to TCI group. The KP score at 6 months in TSI group was significantly higher as well. These results imply that total contact support of medial longitudinal arch is not essential for symptom relief in plantar fasciitis. Previous biomechanical studies mentioned the requisitions of a proper insole for plantar fasciitis. Kogler et al. suggested that orthosis must lower the strain of the foot and the medial surface of the orthosis must stabilize the apical bony structure of the medial longitudinal arch to support it effectively [23, 26] Hsu et al. showed that the plantar fascial stress and peak pressure can be lowered by 14% and 38.9% with optimal insole that supports 54% contact surface of the medial longitudinal arch using finite element method [22]. Three spikes designed in this study were supposed to support more than 50% contact surface of the medial arch.

We can explain why sham orthoses have shown inferior results to custom or prefabricated orthoses with the above-mentioned reasons. Sham orthoses lack sufficient supportive structure for medial longitudinal arch. Landorf et al. compared sham orthosis, prefabricated orthosis,

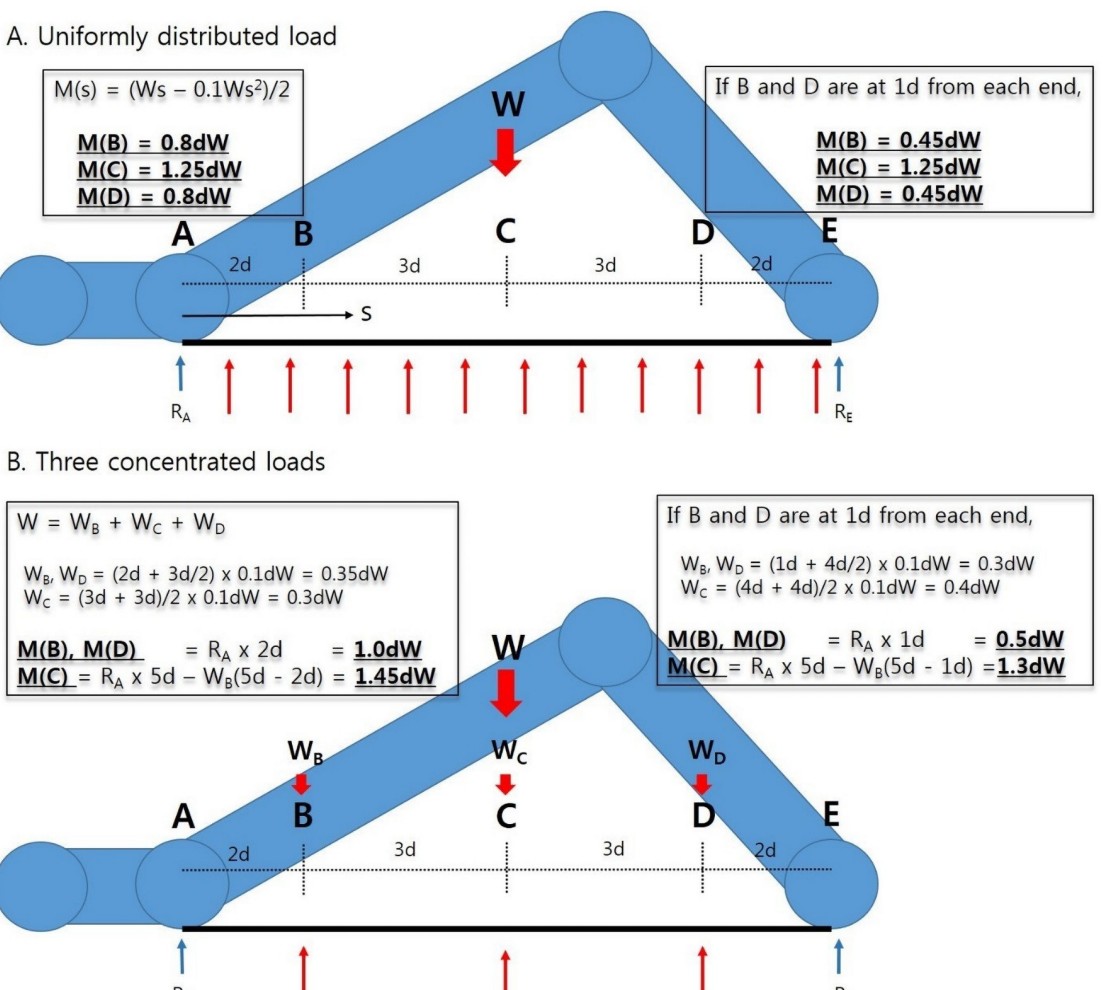

**Fig 3.** A) When we assume that the plantar fascia is simply attached at the ends and the load (W, person's weight) is uniformly distributed, the bending moment at some point (s) can be calculated as formula M(s). Bending moments at each point of B-D would be 0.8dW, 1.25dW, and 0.8dW if we assume that the point B and D are 2d away from each end. B) If multiple three concentrated loads are weighed on, the bending moments at each point of B-D would be 1.0dW, 1.45W, and 1.0dW, respectively. $R_A$, reaction force at point A; $R_E$, reaction force at point E (each $R_A$ and $R_E$ can be calculated as '0.5W' here).

and customized orthosis in their randomized trial [15]. Pain and function scores favored prefabricated and customized orthosis over sham orthosis after 3 months. Bishop et al. compared custom orthosis with new shoes, sham insole with new shoes, and sham insole with patient's regular shoes in a pragmatic randomized controlled study [16]. They concluded that custom orthosis with new shoes was superior in reducing pain and plantar fascia thickness over 12 weeks compared to new shoe alone or sham insole. Similarly, Oliveira et al. showed superior results of custom insole to flat insole with a 6-month follow up [18]. Pain scale and 6-min walk test were significantly better in custom insole group. Bonanno et al. randomized 306 participants from a navy school to flat insole group and prefabricated orthosis group in their prospective study during a 11-week training [27]. Prefabricated orthosis group showed a 34% reduced risk of plantar fasciitis or Achilles tendinopathy. Both groups in the present study showed significant clinical improvements. Semi-rigidity and enough contact surface of the orthosis

**Table 2. Comparisons of clinical outcomes between newly designed TSI and TCI groups.**

| | Baseline | | 6 weeks | | 3 months | | 6 months | | Friedman test | |
|---|---|---|---|---|---|---|---|---|---|---|
| | TSI | TCI | TSI | TCI | TSI | TCI | TSI | TCI | Chi square | P value |
| VAS | 6.2(1.5) | 6.3(1.5) | 3.4(2.1) | 4.8(2.3) | 3.4(1.7) | 3.1(1.5) | 2.4(1.3) | 1.9(1.5) | 61.1 | **.001** |
| AOFAS score | 59.2(13.0) | 56.6(19.3) | 77.5(10.1) | 77.5(6.8) | 77.0(11.2) | 63.5(11.1) | 75.4(11.6) | 77.3(8.4) | 23.7 | **.001** |
| KP score | 47.1(22.3) | 57.1(25.9) | 69.0(22.7) | 66.1(17.4) | **76.1(13.1)** | **59.5(6.2)** | **91.9(4.5)** | **77.8(10.2)** | 25.0 | **.001** |
| FFI | 85.3(30.3) | 89.4(35.6) | 70.6(39.9) | 64.0(21.8) | **49.3(30.8)** | **79.4(17.6)** | 44.0(37.7) | 54.5(31.2) | 33.6 | **.001** |
| FAOS subscales | | | | | | | | | | |
| Pain | 38.7(18.3) | 51.5(25.1) | 63.5(21.6) | 72.1(11.6) | **80.6(13.0)** | **54.9(7.6)** | 74.0(5.8) | 83.7(4.1) | 25.7 | **.001** |
| Symptoms | 78.2(8.4) | 76.6(7.3) | 86.1(9.0) | 85.0(5.9) | **87.7(6.4)** | **70.7(4.8)** | 89.3(4.6) | 86.5(4.3) | 20.6 | **.001** |
| ADL | 74.8(14.4) | 83.0(14.7) | 87.0(10.5) | 86.7(8.2) | 88.7(7.9) | 87.7(4.5) | 85.5(4.5) | 87.0(3.9) | 10.6 | **.014** |
| Sport/Rec | 57.9(13.2) | 69.6(28.5) | 68.3(16.9) | 78.0(8.8) | 69.3(14.1) | 73.4(5.2) | 65.2(7.7) | 67.7(11.6) | 8.8 | **.032** |
| QOL | 24.0(18.4) | 28.6(16.5) | 41.7(24.5) | 53.6(14.9) | 43.8(31.9) | 19.9(10.0) | 57.8(21.1) | 59.4(18.6) | 32.7 | **.001** |
| SF-36 subscales | | | | | | | | | | |
| PCS | 54.1(18.1) | 56.8(15.8) | 60.9(18.7) | 63.3(12.9) | **63.3(16.8)** | **40.8(11.4)** | 64.5(8.8) | 71.8(8.1) | 17.0 | **.001** |
| MCS | 64.3(18.0) | 61.5(9.2) | 78.0(12.7) | 68.6(12.8) | **78.5(8.8)** | **53.1(7.0)** | 79.6(4.2) | 75.6(3.5) | 20.5 | **.001** |

Boldface indicates statistical significance between the groups by post hoc analysis using Mann-Whitney test conducted with a Bonferroni correction (p < .0125). TSI, three spike insole; TCI, total contact insole; VAS, visual analog scale; AOFAS, American Orthopaedic Foot and Ankle Society; KP, Karlsson-Peterson; FFI, foot function index; FAOS, foot and ankle outcome score; ADL, activities of daily living; Sport/Rec, sport and recreational function; QOL, quality of life; SF-36, medical outcomes study short form-36; PCS, physical component summary; MCS, mental component summary.

would have been the sufficient conditions to have a positive effect on plantar fasciitis regardless of the design.

On the other hand, there have been many debates whether prefabricated orthosis is comparable to custom orthosis. Wrobel et al. compared custom foot orthoses, prefabricated foot orthoses, and sham insole on 77 patients in their randomized control study [20]. There were no significant differences among the groups for pain, FFI, and SF-36 for 3 months but the custom orthosis group demonstrated great improvements in spontaneous physical activity compared to prefabricated orthosis and sham insole groups. Xu et al. randomized their 60 patients to customized 3D-printed orthoses and prefabricated orthoses groups [21]. The 3D-printed orthoses group showed superiority in comfort score even though the peak pressure in the hallux and first metatarsal area was significantly higher at initial. Landorf concluded that there were no significant pain and functional improvement between prefabricated insole and custom insole [15]. Baldassin et al. compared the effectiveness of prefabricated and customized foot orthoses [14]. Both groups showed significant pain improvements from baseline to 4- and 8-week but there was no difference between the groups. Instead, they concluded low-cost prefabricated orthosis made up of ethylene vinyl acetate would be the best choice for plantar fasciitis. These debatable results from many previous studies might vary from the shape of prefabricated insole used and subsequent arch support area that could be different in each participant. Provided the medial longitudinal arch of a participant was not sufficiently supported for over 50% surface of arch, the plantar fascial stress and peak pressure would not have been released enough to make obvious clinical results.

Therefore, the clinical results of this study are quite interesting. Newly designed TSI showed not only a non-inferiority but also a significantly exceptional functional improvement and foot-related quality of life during a 6-month follow up. We postulate that these results came from the differences of bending moments in each model. Bigger bending moments in TSI would have added more pressure on plantar fascia and would have led to consequent

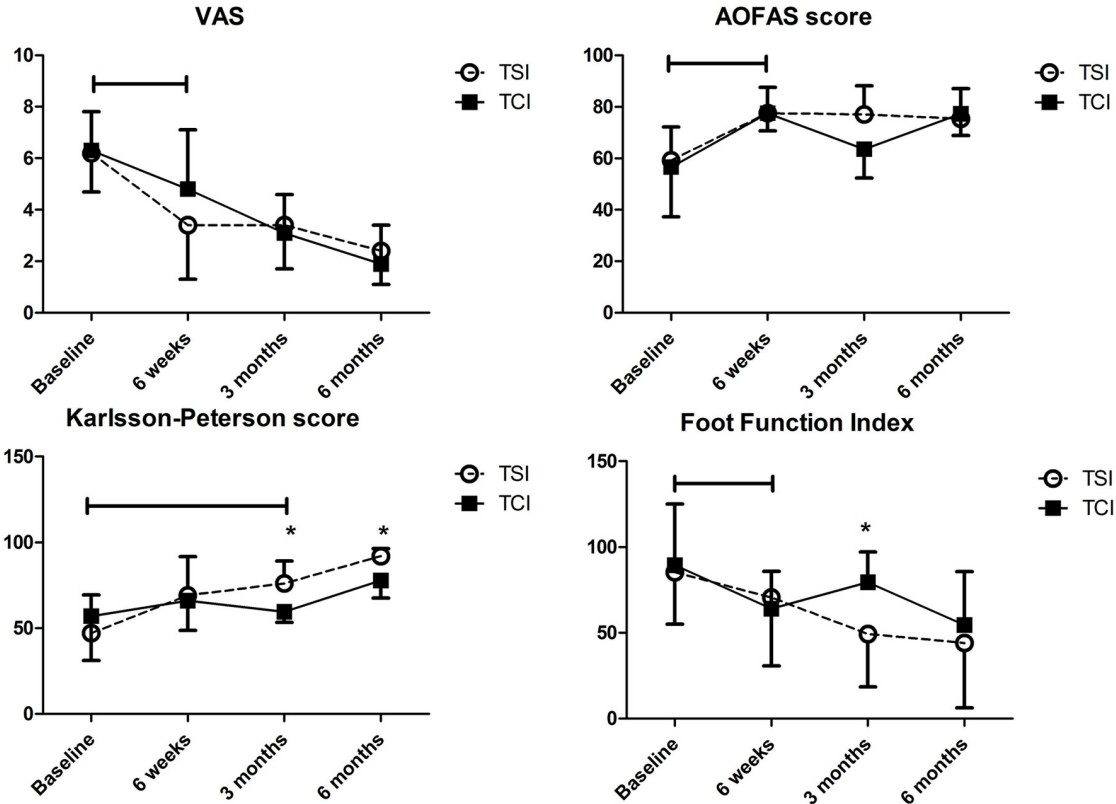

**Fig 4. Clinical outcomes of Visual Analog Scale (VAS), American Orthopaedic Foot and Ankle Society (AOFAS) score, Karlsson-Peterson (KP) score, and Foot Function Index (FFI).** TSI, three spike insole; TCI, total contact insole. * Significant p value < .05 between the groups. Horizontal brackets indicate overall significant improvement of the whole participants from the baseline.

functional improvements. This multiple concentrated load distribution is easy to position according to the length of the patient's plantar fascia. Therefore, mass production of therapeutic insoles would be possible if three spikes are placed along the medial plantar arch of the insole for each size of readymade shoes.

Our study has some limitations. First, the sample size is quite small to draw further differences or identities between the groups in this non-inferiority study model. However, similar demographic distribution to both groups and blinded physician who interpreted the patient outcomes could have offset the possible bias. Although there was a 14.3% loss to follow-up over the course, it was equally allocated between the groups; and the remainders were followed up for a long period of 6 months. Furthermore, KP score, FFI, FAOS, and SF-36 used in this study are validated scores that effectively represent the physical activities and quality of life of the patients. Second, lack of biomechanical study or a reliable plantar pressure measuring system could attenuate our explanation of functional superiority in TSI group. However, different bending moments between the groups according to the above-mentioned structural analysis could explain the reason. Further biomechanical study of this design is needed to support the results of this investigation. Third, 3 persons in 24 participants (12.5%) dropped out because of the inconvenience of the 3D printed insole. The insole was somewhat inevitably rough and bulky to fit inside the patient's own shoes that could have led them to give up. An intention to treat analysis was performed but this could have lessened the strength of this study.

## Conclusion

The present study assessed the effectiveness of a newly designed TSI compared to TCI in refractory plantar fasciitis. We reaffirmed that semi-rigid insole is effective in plantar fasciitis and showed a non-inferiority of TSI based on VAS. In addition, TSI more rapidly restored the pedal function compared to TCI. We assume that high bending moments in TSI model would have added more power on plantar fascia and have led consequent functional improvements. This newly designed TSI model can not only be effective in deriving better clinical outcomes but also be manufactured for popularization due to its reproducible design.

## Supporting information

**S1 Checklist. CONSORT 2010 checklist of information to include when reporting a randomised trial**[*]**.**
(DOC)

**S1 File. File of patient reported clinical outcomes.**
(DOC)

**S2 File.**
(DOC)

**S1 Data.**
(SAV)

## Author Contributions

**Conceptualization:** Dong Woo Shim, Jin Woo Lee.

**Data curation:** Dong Woo Shim, Kyung-Yil Kang.

**Formal analysis:** Seung-Yong Sung, Whan-Yong Chung.

**Funding acquisition:** Sung-Jun Park.

**Investigation:** Dong Woo Shim, Seung-Yong Sung, Whan-Yong Chung.

**Methodology:** Whan-Yong Chung, Jin Woo Lee.

**Project administration:** Jin Woo Lee.

**Resources:** Whan-Yong Chung, Jin Woo Lee.

**Software:** Dong Woo Shim, Seung-Yong Sung.

**Supervision:** Dong Sik Chae.

**Validation:** Dong Sik Chae.

**Visualization:** Dong Woo Shim, Seung-Yong Sung, Whan-Yong Chung.

**Writing – original draft:** Dong Woo Shim.

**Writing – review & editing:** Dong Sik Chae.

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
