## [Decision Letter · Decision Letter 0]

16 Apr 2021

PONE-D-21-05601

Superior pedal function recovery of newly designed three spike insole over total contact insole in refractory plantar fasciitis; a randomized, double-blinded, non-inferiority study

PLOS ONE

Dear Dr. Shim,

Thank you for submitting your manuscript to PLOS ONE. After careful consideration, we feel that it has merit but does not fully meet PLOS ONE’s publication criteria as it currently stands. Therefore, we invite you to submit a revised version of the manuscript that addresses the points raised during the review process.

We look forward to receiving your revised manuscript.

Kind regards,

Bijan Najafi

Academic Editor

PLOS ONE

Journal Requirements:

Thank you for submitting your clinical trial to PLOS ONE and for providing the name of the registry and the registration number. The information in the registry entry suggests that your trial was registered after patient recruitment began. PLOS ONE strongly encourages authors to register all trials before recruiting the first participant in a study.

1) your reasons for your delay in registering this study (after enrolment of participants started);

2) confirmation that all related trials are registered by stating: “The authors confirm that all ongoing and related trials for this drug/intervention are registered.

We suggest you thoroughly copyedit your manuscript for language usage, spelling, and grammar. If you do not know anyone who can help you do this, you may wish to consider employing a professional scientific editing service. 

We note that you have indicated that data from this study are available upon request. PLOS only allows data to be available upon request if there are legal or ethical restrictions on sharing data publicly. For information on unacceptable data access restrictions, please see http://journals.plos.org/plosone/s/data-availability#loc-unacceptable-data-access-restrictions.

4a) If there are ethical or legal restrictions on sharing a de-identified data set, please explain them in detail (e.g., data contain potentially identifying or sensitive patient information) and who has imposed them (e.g., an ethics committee). Please also provide contact information for a data access committee, ethics committee, or other institutional body to which data requests may be sent.

4b) If there are no restrictions, please upload the minimal anonymized data set necessary to replicate your study findings as either Supporting Information files or to a stable, public repository and provide us with the relevant URLs, DOIs, or accession numbers. Please see http://www.bmj.com/content/340/bmj.c181.long for guidelines on how to de-identify and prepare clinical data for publication. For a list of acceptable repositories, please see http://journals.plos.org/plosone/s/data-availability#loc-recommended-repositories.

Please include captions for your Supporting Information files at the end of your manuscript, and update any in-text citations to match accordingly. Please see our Supporting Information guidelines for more information: http://journals.plos.org/plosone/s/supporting-information.

Additional Editor Comments:

Thanks for contributing your original study to PLOS One. It was reviewed by three independent reviewers with complementary expertise in podiatric medicine, biomechanics, and wearables. All reviewers agreed that this study has scientific merits but could be improved further and provided excellent comments. These comments should be addressed before I could recommend acceptance of your manuscript for publication in PLOS One.

Reviewers' comments:

Reviewer's Responses to Questions

**Comments to the Author**

1. Is the manuscript technically sound, and do the data support the conclusions?

Reviewer #1: Yes

Reviewer #2: Yes

Reviewer #3: Yes

2. Has the statistical analysis been performed appropriately and rigorously? 

Reviewer #1: Yes

Reviewer #2: Yes

Reviewer #3: Yes

3. Have the authors made all data underlying the findings in their manuscript fully available?

Reviewer #1: No

Reviewer #2: No

Reviewer #3: Yes

4. Is the manuscript presented in an intelligible fashion and written in standard English?

Reviewer #1: Yes

Reviewer #2: No

Reviewer #3: Yes

5. Review Comments to the Author

Reviewer #1: A randomized controlled noninferiorty clinical trial was conducted to compare VAS scores of the total contact sole (TCS) with a newly designed three-spike insole (TSI). The study results indicated that the experimental design was noninferior to the standard insole design. Improvements over time were noted in both groups.

Minor revisions:

1- Abstract: Rephrase to improve clarity: “Twenty-eight patients were randomized with equal allocation to either TSI or TCI.”

2- Express numerical p-values more precisely, rather than p < 0.0xx.

3- Line 112: Specify the block size used in the randomization.

4- Line 162: Use standard phrasing: “Twenty-eight patients were randomized, with equal probability, to two groups.”

5- For normally distributed variables, summarize with mean and standard deviation. For nonparametric variables, summarize using median, first, and third quartiles.

6- Indicate if any adverse events occurred during the course of the study.

Reviewer #2: The authors conducted a randomized controlled trial comparing the effect of newly designed insole and total contact insole on plantar fasciitis symptoms. Overall, this study seems well-conducted with reasonable designs and results. I have a few comments.

Introduction

In the first paragraph, please include some data from Korea, if available. For example, the prevalence of plantar fasciitis in Korea, economic cost, etc.

The primary aim is to test new spike insole, and yet it is not very clear to me what are the problems or limitations with conventional insoles. Is the problem just the cost and the duration of fabrication? Also, why did the authors come up with the spike, that is why did the authors believe the spiky design would address the limitations of conventional insoles?

The purpose statement needs to be added.

Methods

In the figure 1, please add a side view photo so that readers can see the height of spikes.

Also, by any chance, did the authors have any data from patients' perceive satisfaction for the new insole? If so, please add the results.

Why did the authors use Mann-Whitney? All the variables were non-parametric? Please confirm.

Results

The figure 2 is difficult to understand. What exactly the asterisk represent? Significant difference in the spike insole group or total contact insole? Do the asterisks represent time x group effect or just time effect?

No new figure should be introduced in discussion so please move figure 4 to results section, and describe the results in this section.

Discussion and conclusion seems to be fine.

Reviewer #3: The authors present a well designed and written study evaluating the efficacy of a novel “3-spike” design shoe insert for the treatment of plantar fasciitis recalcitrant to prior treatment.

To evaluate efficacy the authors enrolled 14 patients to each arm of a non-inferiority comparison between standard orthotic device and novel interventional device. Patients were tracked for 6 months and evaluated using multiple validated subjective scoring systems.

The interventional inserts were created using molds of the patient’s feet, and were 3-D printed to match the patient's weightbearing medial arch.

Utilizing prior experience and observational information regarding the nature of the mold the length and position of the medial arch band of the plantar fascia was interpreted and insoles were either made to be A: load bearing along the full length (Control), or B: fitted with three 2cm diameter elevated “spikes” at center point and distal/proximal along the medial band.

Ultimately the authors found the patient's pain and functional capacity were normalized faster utilizing novel design as compared to standard of care design (control). In this regard the author’s conclude this novel technique to be at minimum non-inferior and likely superior to standard of care.

The authors chose to craft booth insoles using similar methodology to control for variations in design other than the interventional modification. The authors state their believed limitations of this study regarding limited treatment group size and lack of biomechanical or pressure data to support hypothesis of function and loading on novel device. Despite these limitations the authors have designed and executed a well-developed study with novel implications for clinical management of a frequently encountered condition.

I recommend this article for acceptance with minor grammatical revision and look forward to future work from these authors in this field.

6. PLOS authors have the option to publish the peer review history of their article (what does this mean?). If published, this will include your full peer review and any attached files.

Reviewer #1: No

Reviewer #2: No

Reviewer #3: No

---

## [Author Response · Author response to Decision Letter 0]

9 May 2021

 → Thank you for your pointing out. We reaffirmed that we have revised the format as introduced and met the requirements. 

2. Thank you for submitting your clinical trial to PLOS ONE and for providing the name of the registry and the registration number. The information in the registry entry suggests that your trial was registered after patient recruitment began. PLOS ONE strongly encourages authors to register all trials before recruiting the first participant in a study.

1) your reasons for your delay in registering this study (after enrolment of participants started);

→ Thank you for your pertinent comment. We commented the reason of delay. Actually, we had no idea of the public registration at the beginning of this study. (Line 98-99)

2) confirmation that all related trials are registered by stating: “The authors confirm that all ongoing and related trials for this drug/intervention are registered.

→ Thank you for your comment. We added the sentence right after the above mentioned amendment. (Line 99-101)

→ Thank you for your suggestion. We have made our manuscript edited once again via BIENTrans and uploaded the certificate as well.

 4a) If there are ethical or legal restrictions on sharing a de-identified data set, please explain them in detail (e.g., data contain potentially identifying or sensitive patient information) and who has imposed them (e.g., an ethics committee). Please also provide contact information for a data access committee, ethics committee, or other institutional body to which data requests may be sent.

 4b) If there are no restrictions, please upload the minimal anonymized data set necessary to replicate your study findings as either Supporting Information files or to a stable, public repository and provide us with the relevant URLs, DOIs, or accession numbers. Please see http://www.bmj.com/content/340/bmj.c181.long for guidelines on how to de-identify and prepare clinical data for publication. For a list of acceptable repositories, please see http://journals.plos.org/plosone/s/data-availability#loc-recommended-repositories.

→ Thank you for your reminder. We uploaded our data set as Supporting Information and have mentioned it in the revised cover letter. 

 → Thank you for your guidance. We have added a sentence at the end of the manuscript. 

→ Thank you for your reminder. We have checked listed references once again and they are complete and correct.

Additional Editor Comments:

Thanks for contributing your original study to PLOS One. It was reviewed by three independent reviewers with complementary expertise in podiatric medicine, biomechanics, and wearables. All reviewers agreed that this study has scientific merits but could be improved further and provided excellent comments. These comments should be addressed before I could recommend acceptance of your manuscript for publication in PLOS One.

Reviewers' comments:

5. Review Comments to the Author

Reviewer #1: A randomized controlled noninferiorty clinical trial was conducted to compare VAS scores of the total contact sole (TCS) with a newly designed three-spike insole (TSI). The study results indicated that the experimental design was noninferior to the standard insole design. Improvements over time were noted in both groups.

Minor revisions:

1- Abstract: Rephrase to improve clarity: “Twenty-eight patients were randomized with equal allocation to either TSI or TCI.”

→ We thank you for your clarification. We have amended it as you have guided.

2- Express numerical p-values more precisely, rather than p < 0.0xx.

→ We thank you for these very important comments. We have proposed more precise ones as you have commented. (Line 51, 53, 190, 192, 195)

3- Line 112: Specify the block size used in the randomization.

→ We thank you for these pertinent comments. We used four block randomization and added it as you have recommended. (Line 103)

4- Line 162: Use standard phrasing: “Twenty-eight patients were randomized, with equal probability, to two groups.”

→ We thank you for your clarification. We revised it as you have recommended. (Line 154)

5- For normally distributed variables, summarize with mean and standard deviation. For nonparametric variables, summarize using median, first, and third quartiles.

→ We thank you for this perceptive comment. We added standard deviation of each value right after mean value. (Line 158-160)

6- Indicate if any adverse events occurred during the course of the study.

→ Thank you for your comment. There was no adverse effect except for the dropped cases as we described in CONSORT flowchart. We specified the absence of adverse event (Line 197) 

Reviewer #2: The authors conducted a randomized controlled trial comparing the effect of newly designed insole and total contact insole on plantar fasciitis symptoms. Overall, this study seems well-conducted with reasonable designs and results. I have a few comments.

Introduction

In the first paragraph, please include some data from Korea, if available. For example, the prevalence of plantar fasciitis in Korea, economic cost, etc.

→ Thank you for your comment. Unfortunately, there was no study on prevalence of plantar fasciitis in Korea. However, we added the information of the economic cost in Korea in accordance with National Health Insurance Service. (Line 63-65)

The primary aim is to test new spike insole, and yet it is not very clear to me what are the problems or limitations with conventional insoles. Is the problem just the cost and the duration of fabrication? Also, why did the authors come up with the spike, that is why did the authors believe the spiky design would address the limitations of conventional insoles?

The purpose statement needs to be added.

→ Thank you for your pertinent question. As you have mentioned again, the conventional insoles are quite expensive and take time for customized production. However, we focused on that partially arch supporting structured insole could have a comparable load distribution effect on plantar fascia. Moreover, we postulated that if it has stimulation effect on both ends of fascia, the fascia releasing effect could be maximized. That is why we initially designed the three-spike insole. This design has the merit over conventional TCI on that the three spikes can be easily located uniformly along the plantar fascia. It can be applied to the insoles commonly used in the market with the effect of fascial stretching, saving time and money compared to a TCI and thus allowing mass production of therapeutic insole with the above design. However, the purpose does not contain the exact content as you pointed. We revised it. (Line 82)

Methods

In the figure 1, please add a side view photo so that readers can see the height of spikes.

→ Thank you for your recommendation. We added the lateral view of each insole in Fig 1. 

Also, by any chance, did the authors have any data from patients' perceive satisfaction for the new insole? If so, please add the results.

→ Thank you for your question. Unfortunately, we did not check the patient reported satisfaction scale for each insole. 

Why did the authors use Mann-Whitney? All the variables were non-parametric? Please confirm.

→ We thank you for your perceptive comment. We used Mann-Whitney and Fisher’s exact test conservatively because many but not all values showed non-parametric distribution.

Results

The figure 2 is difficult to understand. What exactly the asterisk represent? Significant difference in the spike insole group or total contact insole? Do the asterisks represent time x group effect or just time effect?

→ We are grateful for you pointing out this ambiguity. We guess you are pointing Figure 3. Captions might have insufficient explanation. Individual asterisk points significant difference between the groups. And the bar indicates the overall improvement from baseline in both groups. We removed the asterisk over the bar since it was confusing. 

No new figure should be introduced in discussion so please move figure 4 to results section, and describe the results in this section.

→ Thank you for your recommendation. We stated the calculation of bending moments at the end of the Material and Methods section (Line 127-128) and moved the figure and results to the Result section. (Line 169-180)

Discussion and conclusion seems to be fine.

Reviewer #3: The authors present a well designed and written study evaluating the efficacy of a novel “3-spike” design shoe insert for the treatment of plantar fasciitis recalcitrant to prior treatment.

To evaluate efficacy the authors enrolled 14 patients to each arm of a non-inferiority comparison between standard orthotic device and novel interventional device. Patients were tracked for 6 months and evaluated using multiple validated subjective scoring systems.

The interventional inserts were created using molds of the patient’s feet, and were 3-D printed to match the patient's weightbearing medial arch.

Utilizing prior experience and observational information regarding the nature of the mold the length and position of the medial arch band of the plantar fascia was interpreted and insoles were either made to be A: load bearing along the full length (Control), or B: fitted with three 2cm diameter elevated “spikes” at center point and distal/proximal along the medial band.

Ultimately the authors found the patient's pain and functional capacity were normalized faster utilizing novel design as compared to standard of care design (control). In this regard the author’s conclude this novel technique to be at minimum non-inferior and likely superior to standard of care.

The authors chose to craft booth insoles using similar methodology to control for variations in design other than the interventional modification. The authors state their believed limitations of this study regarding limited treatment group size and lack of biomechanical or pressure data to support hypothesis of function and loading on novel device. Despite these limitations the authors have designed and executed a well-developed study with novel implications for clinical management of a frequently encountered condition.

I recommend this article for acceptance with minor grammatical revision and look forward to future work from these authors in this field.

→ We are sincerely grateful for your compliments. We are preparing the subsequent large cohort study and hope to introduce it to you in the future.

---

## [Decision Letter · Decision Letter 1]

9 Jul 2021

Superior pedal function recovery of newly designed three spike insole over total contact insole in refractory plantar fasciitis; a randomized, double-blinded, non-inferiority study

PONE-D-21-05601R1

Dear Dr. Shim,

We’re pleased to inform you that your manuscript has been judged scientifically suitable for publication and will be formally accepted for publication once it meets all outstanding technical requirements.

Kind regards,

Bijan Najafi

Academic Editor

PLOS ONE

Additional Editor Comments (optional):

Thanks for your efforts in addressing the initial voiced concerns raised by reviewers. One of the reviewers voiced additional concern. However, since this concern is minor, to avoid further delay in the review process I recommend acceptance of your manuscript with condition that this concern is addressed in revising your proof.

Reviewers' comments:

Reviewer's Responses to Questions

**Comments to the Author**

1. If the authors have adequately addressed your comments raised in a previous round of review and you feel that this manuscript is now acceptable for publication, you may indicate that here to bypass the “Comments to the Author” section, enter your conflict of interest statement in the “Confidential to Editor” section, and submit your "Accept" recommendation.

Reviewer #1: (No Response)

Reviewer #2: All comments have been addressed

Reviewer #3: All comments have been addressed

2. Is the manuscript technically sound, and do the data support the conclusions?

Reviewer #1: Yes

Reviewer #2: Yes

Reviewer #3: Yes

3. Has the statistical analysis been performed appropriately and rigorously? 

Reviewer #1: Yes

Reviewer #2: Yes

Reviewer #3: Yes

4. Have the authors made all data underlying the findings in their manuscript fully available?

Reviewer #1: No

Reviewer #2: No

Reviewer #3: Yes

5. Is the manuscript presented in an intelligible fashion and written in standard English?

Reviewer #1: Yes

Reviewer #2: Yes

Reviewer #3: Yes

6. Review Comments to the Author

Reviewer #1: Line 151: Clarify how the non-inferiority test was used. Specifically state the hypothesis it tested.

Thoroughly proofread the manuscript to ensure proper grammar.

Reviewer #2: I thank the authors for addressing my comments. The manuscript looks good. I have no additional comments.

Reviewer #3: Thank you for contributing your original study to PLOS One & making revisions as described by reviewers and editors. After further review of revised materials I recommend acceptance of your manuscript for publication in PLOS One, and look forward to further work from this group.

7. PLOS authors have the option to publish the peer review history of their article (what does this mean?). If published, this will include your full peer review and any attached files.

Reviewer #1: No

Reviewer #2: No

Reviewer #3: No

---

## [Editor Report · Acceptance letter]

15 Jul 2021

PONE-D-21-05601R1 

Superior pedal function recovery of newly designed three spike insole over total contact insole in refractory plantar fasciitis; a randomized, double-blinded, non-inferiority study 

Dear Dr. Shim:

I'm pleased to inform you that your manuscript has been deemed suitable for publication in PLOS ONE. Congratulations! Your manuscript is now with our production department. 

Kind regards, 

on behalf of

Dr. Bijan Najafi 

Academic Editor

PLOS ONE